# The Dose Response for Sprint Interval Training Interventions May Affect the Time Course of Aerobic Training Adaptations

**DOI:** 10.3390/sports7040085

**Published:** 2019-04-10

**Authors:** Dominic O’Connor, John K. Malone

**Affiliations:** 1O’Brien Centre for Science, University College Dublin, D04 V1W8 Dublin, Ireland; 2School of Social and Health Sciences, Abertay University, DD1 1HG Dundee, Scotland, UK; j.malone@abertay.ac.uk

**Keywords:** Inactive populations, cardiorespiratory fitness, high intensity interval training, psychological indices

## Abstract

Low vs. high volume sprint-interval training (SIT) sessions have shown similar physiological benefits after 8 weeks. However, the dose response and residual effects of shorter SIT bouts (<10 s) are unknown. Following a 6-wk control period, 13 healthy inactive males were assigned to a low dose (LDG: *n* = 7) or high dose (HDG: *n* = 6) supervised 6-wk intervention: ×2/wk of SIT (LDG = 2 sets of 5 × 6 s ON: 18 s OFF bouts; HDG = 4–6 sets); ×1/wk resistance training (3 exercises at 3 × 10 reps). Outcome measures were tested pre and post control (baseline (BL) 1 and 2), 72 h post (0POST), and 3-wk post (3POST) intervention. At 0POST, peak oxygen uptake (VO_2peak_) increased in the LDG (+16%) and HDG (+11%) vs. BL 2, with no differences between groups (*p* = 0.381). At 3POST, VO_2peak_ was different between LDG (−11%) and HDG (+3%) vs. 0POST. Positive responses for the intervention’s perceived enjoyment (PE) and rate of perceived exertion (RPE) were found for both groups. Blood pressure, blood lipids, or body composition were not different between groups at any time point. Conclusion: LDG and HDG significantly improved VO_2peak_ at 0POST. However, findings at 3POST suggest compromised VO_2peak_ at 0POST in the HDG due to the delayed time course of adaptations. These findings should be considered when implementing high-dose SIT protocols for non-athletic populations.

## 1. Introduction

Despite the well documented benefits of physical activity, over 30% of adults worldwide are physically inactive [1]. ‘Lack of time’ is commonly reported as a barrier to physical activity [2], and may in part, explain the low participation and adherence to current exercise guidelines, which are based on time-intensive moderate intensity continuous training (MICT). Accordingly, time-efficient paradigms, such as high intensity interval training (HIIT) and sprint interval training (SIT), have been developed. While both paradigms are associated with a plethora of health benefits, including increased cardiorespiratory fitness [3], the potential time commitment can still be considerable. For example, HIIT protocols involving 1 to 4 min bouts of intense exercise within a session, including a warm up, recovery periods, and a cool down, last approximately 20 to 25 min [4]. Likewise, traditional SIT protocols, such as the protocol by Burgomaster et al. [5], typically utilize multiple 30 s “all-out” efforts with a 1:8 work-to-rest ratio (W:R). Conversely, SIT protocols involving shorter work bouts (<10 s) may utilize considerably shorter recovery periods between bouts, e.g., 8 s ON:12 s OFF [6], making them potentially more time efficient, assuming prior consideration of appropriate warm-up/cool-down. 

The potency of SIT for inducing health adaptations and improving cardiorespiratory fitness is known. For example, Gillen et al. [7] showed that 3 × 20 s sprints separated by 2 min of rest (1:6 W:R), ×3/wk for 6 wks improved peak oxygen uptake (VO_2peak_) by 12% in healthy overweight/obese populations. However, despite its potency, uptake and adherence to SIT in inactive populations can be limited by its challenging “all-out” nature, which can induce feelings of nausea and dizziness [8]. Despite this, some studies have shown higher intensity protocols to be perceived as more enjoyable than MICT [9]. In addition, reducing overall session volume load by using protocols involving fewer or shorter sprints may enhance uptake and adherence [10].

Recently, Logan et al. [11] demonstrated that reducing session volume may not diminish acquired training adaptations. In their study in adolescents, they compared 1 set vs. 5 sets of 4 × 20 s bouts interspersed with 10 s passive recovery (2:1 W:R) over 8 weeks and reported no significant differences between groups for health markers, notably cardiorespiratory fitness and body composition. However, despite this finding, issues remain regarding the application of common SIT protocols in inactive populations. First, longer bouts of SIT (≥20 s) may still pose a considerable physiological challenge. Second, differences in training loads may impart differential adaptation time-course kinetics; that is, an extended ‘recovery’ period may be required to realize the true magnitude of adaptations to high-dose SIT owing to possible temporary physiological suppression due to overreaching post intervention and delayed training adaptations [12]. Third, the existing literature often fails to consider the paradigm shift towards concurrent training, i.e., in combination with resistance training (RT), in non-athletic populations due to its combined physiological benefits. Currently, there is little emphasis on its use in research studies [13] despite being recommended in current exercise guidelines. 

To help increase the uptake and long-term adherence to habitual physical activity and overcome the issues outlined, there needs to be an emphasis on implementing appropriate exercise protocols that are easily administered, time-efficient, and tolerable. To address the issues outlined, we aimed to investigate the dose-response of two short-bout (6 s) SIT/RT concurrent exercise interventions in previously inactive adult males to determine whether: (1) The low dose SIT group (LDG) experienced similar physiological adaptations to the high dose SIT group (HDG); and (2) a training cessation period post intervention demonstrated evidence of super-compensation in the HDG. We hypothesized that: (1) No significant differences would exist between groups post-intervention; and (2) the HDG would display significantly greater adaptations compared to the LDG at 3 weeks post.

## 2. Material and Methods

### 2.1. Experimental Design

A between-groups design comprising a 6-wk control period (maintenance of habitual and dietary lifestyle—wks 1–6), a 6-wk supervised training intervention (3×/wk, 2 × SIT/1 × RT) (wks 7–12), and a 3-wk training cessation period (wks 13–15). Physiological outcome measures were recorded at 4 separate time points: Start of control period (BL1); post 6-wk control period (BL2); post 6-wk training intervention (0POST); 3-weeks post intervention (3POST). Psychological outcome measures were recorded for SIT sessions during the 6 wk training intervention.

### 2.2. Participants

Following institutional ethical approval (SHS_T_2015-16_887) 19 healthy inactive males volunteered to participate. However, six participants withdrew prior to completion (illness, *n* = 2; time commitments, *n* = 4) leaving a total of 13 participants included for final analysis (one participant in the LDG was unable to complete testing at 3POST, therefore, for all outcome variables, an *n* = 6 for both groups is reported at this time point). Participants were allocated to a training group using a random, stratified approach based on baseline VO_2peak_, into LDG (*n* = 7; age: 35 ± 7.1 yr; HT: 176.8 ± 5.7 cm; BM: 95.4 ± 27.8 kg; BMI: 30.5 ± 7.5 kg·m^−2^) and HDG (*n* = 6; age 38 ± 7.5 yr; HT: 177.5 ± 3.8 cm; BM 86.4 ± 7.0 kg; BMI: 27.4 ± 2.1 kg·m^−2^). Females were not considered to avoid the influence of menstruation, e.g., subjective responses to exercise [14]. Inclusion criteria included: (1) Inactive lifestyle based on self-reported habitual physical activity of ≤1 h of structured exercise per wk [7]; (2) answered NO to all questions on the physical activity readiness questionnaire (PAR-Q); and (3) no history of cardiovascular, metabolic, or hormonal disease. Participants were fully informed of all experimental procedures before providing written informed consent. 

### 2.3. Control Period (BL1 and 2—Wk 1–6)

This period began immediately after BL1 and ended after BL2 in wk 6. Participants were instructed to maintain their normal habitual dietary and activity lifestyle during this period. 

### 2.4. Training Intervention (Wk 7–12)

Where possible, SIT sessions were completed on Mondays and Fridays, with the RT session on Wednesdays. In all cases, a minimum of 48 h separated sessions. 

*Sprint Interval Training (SIT):* Each set consisted of five bouts of “all out” maximal exercise against a resistance of 7.5% body weight [15] on a cycle ergometer (Ergomedic 874E, Monark, Vansbro, Sweden), with 2 min recovery between sets. Resistance dropped at 100 rev·min^−1^ and each bout lasted 6 s, interspersed with 18 s of unloaded cycling at 40 rev·min^−1^ (i.e., 1 set = 5 × 6:18 s). The LDG (*n* = 7) consisted of 2 sets per session, and the HDG (*n* = 6) consisted of 4 (wks 1–2), 5 (wks 3–4), and 6 (wks 5–6) sets per session. Participants received strong verbal encouragement during bouts. RPE and PE were assessed at the end of each SIT session using a Borg 15-point scale, and an arbitrary scale that ranged from 0 (not enjoyable at all) to 9 (very enjoyable), respectively [16]. A 2-min warm-up and cool-down of unloaded cycling at 60 rpm was completed before and after each session. 

*Resistance Training (RT):* Participants’ 10 repetition maximum (10-RM) was determined for the chest press, seated row, and leg press machines to enable intensities of training to be administered. Participants completed a warm-up of 10 repetitions at 50% 10 R-M on each of the exercises prior to completing 3 sets (2 sets of 10 repetitions with the last set completed to failure) for each exercise [11], using a tempo of 2–0–2 s for the concentric–isometric–eccentric phases at 10-RM intensity. 

### 2.5. Training Cessation Period (Wk 13–15)

Upon intervention cessation, participants were instructed to revert to their previous sedentary habitual lifestyle similar to their control period and made aware of the importance of adherence to the study. All participants verbally reported that they did revert to their pre-intervention habitual lifestyles prior to the 3POST. 

### 2.6. Testing Procedures 

On a separate day to BL1, participants attended a familiarization session, which included completing one set of the SIT protocol and familiarization with all other testing procedures. 

#### 2.6.1. Baseline 1 (BL1)

Prior to BL1, participants completed a detailed food and activity log for the 24 h period prior to testing and were instructed to avoid strenuous exercise during this time. Participants were required to refrain from alcohol consumption for 24 h or caffeine for 9 h and were instructed to consume 1 L of water ad libitum over the final 2 h period prior to attendance and advised to empty their bladders prior to testing. Participants reported to the human performance laboratory in a fasted state (minimum of 9 h). 

*Anthropometry:* Participants removed footwear, socks, and all jewellery, and height (to nearest 0.1 cm) and body mass (to nearest 0.1 kg) were recorded using a portable standing stadiometer (Seca: 213, Seca, Chino, CA, United States) and a calibrated scale (Tanita: MC-780, Tanita Inc., Tokyo, Japan), respectively. Segmental fat mass (%) and muscle mass (%) were measured via bioelectrical impedance analysis (Tanita: MC-780, Tanita Inc., Tokyo, Japan). Right leg girth (RLG) measurement was taken in accordance with the American College of Sports Medicine (ACSM) guidelines [17]. Without compressing subcutaneous adipose tissue, a flexible Rollflix tape measure (Anthropometric tape, HaB direct, Southam, Warwickshire, UK) was placed onto the surface of the skin with the participant standing with one leg on a bench with the knee flexed at 90 degrees. The measure was taken midway between the inguinal crease and the proximal border of the patella, perpendicular to the long axis. Measures were duplicated, and retested if not within 5 mm [17].

*Blood Pressure:* Participants remained seated quietly in the Fowler position for 10-min prior to 3 separate measurements of blood pressure using an automated blood pressure monitor (Omron: M3, Omron, Kyoto, Japan) measured on the non-dominant arm. The average of three measurements was recorded [18]. 

*Blood Analysis:* Three separate 15 µL capillary blood samples were collected under sterile conditions via a pin prick of the fingertip and analysed for total cholesterol (TC), high density lipoprotein (HDL), and triglycerides (TG) using a portable blood lipid analyser (Cardiocheck: PA, Polymer Technology Systems, Indianapolis, IN, United States). 

*Standardized Breakfast:* Following blood sampling, participants consumed 30 g cornflakes with 75 mL semi-skimmed milk, 42 g cereal bar, and 200 mL of an isotonic sports drink (384 kcal carbohydrate 70.2 g, fat 7.7 g, protein 8.4 g).

*Incremental Exercise Test:* Participants performed an incremental peak oxygen uptake (VO_2peak_) exercise test on a cycle ergometer (Ergomedic 874E, Monark, Vansbro, Sweden). Participants cycled at 60 W and 60 rev·min^−1^ for 1 min, before the workload was increased by 30 W per minute until volitional exhaustion. Expired gases were measured breath-by-breath using an online gas analyser (Metalyzer3B gas analyzer; Cortex, Leipzig, Germany). Heart rate (HR) was recorded at the end of each stage, using wireless telemetry (Polar Electro, Kempele, Finland), with time to exhaustion (TTE) recorded to the nearest 1 s.

#### 2.6.2. BL2, 0POST, and 3POST

All procedures for subsequent testing sessions were replicated exactly as for BL1. Participants were instructed to stringently replicate their food and activity logs for the 24 h prior to testing and were made aware of the importance of this for the validity of the data. To control for circadian rhythm, sessions were performed at the same time of day (±1 h) [19]. 

### 2.7. Data and Statistical Analysis

All data are presented as mean ± SD. Prior to conducting parametric tests, a Shapiro-Wilks test was carried out to ensure data was normally distributed. Data were analysed using a 2 × 4 repeated analysis of variance (ANOVA), with the between-factor “group” (i.e., LDG vs. HDG) and the within-factor “time” (i.e., BL1, BL2, 0POST, 3POST) with alpha ≤0.05 using SPSS Version 23.0 software (SPSS Inc., Chicago, IL, USA). Significant interactions and main effects were analysed using Bonferonni post-hoc test. Comparisons in RPE and PE data between groups were made using an independent sample t-test. Within group comparisons were made using a paired sample t-test. Cohen’s effect size was determined at: 0.1–0.3, small effect; 0.3–0.5, moderate effect; and 0.5–0.7, large effect [20].

## 3. Results

### 3.1. Indices of Cardiorespiratory Fitness

*VO_2peak_:* As shown in Figure 1A, there was a significant interaction for group × time (*p* = 0.013), and main effects of group (*p* < 0.001) and time (*p* < 0.001). There were no significant differences between BL1 and BL2 for either group (LDG: 31.9 ± 8.4 vs. 30.8 ± 8.0 mL·kg^−1^·min^−1^, *p* = 0.23; HDG: 34.0 ± 2.5 vs. 34.4 ± 3.1 mL·kg^−1^·min^−1^, *p* = 0.53). Both training groups had significantly improved VO_2peak_ values at 0POST (LDG: +16%, 32.3 ± 7.6 vs. 37.6 ± 10.0 mL·kg^−1^·min^−1^, *p* < 0.001, d = 0.7; HDG: +11%, 34.4 ± 3.1 mL·kg^−1^·min^−1^ vs. 38.1 ± 3.8 mL·kg^−1^·min^−1^, *p* < 0.001, d = 1.2) with no significant difference between groups (*p* = 0.381, d = 0.1). At 3POST, VO_2peak_ values were significantly different between groups (*p* = 0.002, d = 0.6). VO_2peak_ decreased by 11% in the LDG (37.6 ± 10.0 vs. 33.4 ± 9.4 mL·kg^−1^·min^−1^, *p* = 0.03, d = −0.5) with no change in the HDG (+3%, 38.1 ± 3.8 vs. 39.2 ± 4.2 mL·kg^−1^·min^−1^, *p* = 0.09, d = 0.2) (individual data shown in Figure 2). 

*Time to Exhaustion (TTE):* As shown in Figure 1B, there was no significant interaction for group × time (*p* = 0.339). There were no significant main effects of group (*p* = 0.659) or time (*p* = 0.139). At 0POST, TTE did not significantly increase in either group (*p* = 0.130) (LDG: +9%, 433 ± 41 vs. 472 ± 6 s; HDG: +7%, 443 ± 79 vs. 474 ± 90 s) with no difference between groups (*p* > 0.05, d = 0.1). At 3POST, TTE was not significantly different (*p* = 0.214) (LDG: −9%, 472 ± 62 vs. 428 ± 83 s, HDG: +1%, 474 ± 89 vs. 479 ± 90 s) with no difference between groups (*p* = 0.233, d = 0.6). 

### 3.2. Psychological Responses

*Ratings of Perceived Exertion (RPE):* As shown in Figure 3A, there was no significant difference between groups after session 1 (LDG; 18 ± 1, HDG; 17 ± 1, *p* = 0.127, d = 0.2) or after session 12 (LDG; 15 ± 1, HDG; 14 ± 1, *p* = 0.390, d = 0.2). After the intervention, RPE significantly decreased after session 12 in both LDG (18 ± 1 to 15 ± 2, *p* = 0.017, d = 1.3) and HDG (17 ± 1 to 14 ± 2, *p* = 0.023, d = 1.9).

*Perceived Enjoyment (PE):* As shown in Figure 3B, there was no significant difference between groups after session 1 (LDG; 5 ± 1, HDG; 6 ± 1 AU, *p* = 0.127, d = 0.2) or after session 12 (LDG; 7 ± 1, HDG; 7 ± 1 AU, *p* = 0.390, d = 0.2). After the intervention, PE significantly improved after session 12 in both LDG (5 ± 2 to 7 ± 1 AU, *p* = 0.039, d = 1.3) and HDG (6 ± 2 to 7 ± 1 AU, *p* < 0.001, d = 1.4). 

### 3.3. Blood Lipid Profile

*High Density Lipoproteins (HDL).* There was no significant interaction for group × time (*p* = 0.281) and main effects for group (*p* = 0.816) or time (*p* = 0.097). There were no significant difference in HDL at 0POST, (LDG: +3%, 50 ± 10 vs. 51 ± 14 mg/dL: HDG: +8%, 44 ± 6 vs. 48 ± 6 mg/dL, *p* < 0.05, d = 0.3) or at 3POST (LDG: −8%, 51 ± 14 vs. 47 ± 4 mg/dL; HDG: +5%, 48 ± 6 vs. 50 ± 13 mg/dL, *p* < 0.05, d = 0.3). 

*Total Cholesterol (TC):* There was no significant interaction for group × time (*p* = 0.658) or effect for group (*p* = 0.784). There was a significant main effect for time (*p* = 0.009). TC significantly increased between BL1 and BL2 for both groups (LDG: 180 ± 42 vs. 201 ± 30 mg/dL, *p* = 0.035; HDG: 173 ± 31 vs. 185 ± 57 mg/dL, *p* = 0.041). At 0POST, TC had decreased significantly. TC had decreased by 16% and 7% in the LDG (201 ± 30 vs. 169 ± 37 mg/dL) and the HDG (185 ± 57 vs. 172 ± 51 mg/dL) groups, respectively. At 3POST, TC significantly increased in both groups with a mean increase in TC of 11% and 6% in LDG (169 ± 37 vs. 188 ± 42 mg/dL, *p* = 0.02) and HDG (172 ± 51 vs. 182 ± 45 mg/dL, *p* = 0.033), respectively.

*Triglycerides (TG):* There was no significant interaction for group × time (*p* = 0.658) or main effect for group (*p* = 0.784). There was a significant main effect for time (*p* = 0.016). TG significantly increased between BL1 and BL2 for both groups (LDG: 107 ± 44 vs. 134 ± 66 mg/dL, HDG: 60 ± 14 vs. 109 ± 55 mg/dL). At 0POST, there were no significant changes in both groups (LDG −26%, 143 ± 66 vs. 99 ± 42 mg/dL; HDG: −15%, 109 ± 55 vs. 92 ± 39 mg/dL). At 3POST, there was no significant change in TG in both groups (LDG: +1%, 99 ± 42 vs. 100 ± 21 mg/dL; HDG: +14%, 92 ± 39 vs. 105 ± 32 mg/dL).

### 3.4. Anthropometric Indices

For all anthropometric indices, there were no significant differences between BL1 and BL2 for any variables for either group. Aside from a significant interaction for time—at 0POST, RLG was greater than BL2 in the LDG—there were no significant differences between groups for any of the variables at any time point (see Table 1). 

### 3.5. Blood Pressure

*Systolic Blood Pressure (SBP):* There was no significant interaction for group × time (*p* = 0.596), or main effect for group (*p* = 0.643) or time (*p* = 0.148). SBP did not significantly change in either group at 0POST (LDG: +2%, 133 ± 13 vs. 131 ± 13 mmHg; HDG: +2%, 130 ± 10 vs. 128 ± 9 mmHg) or at 3POST (LDG: −2%, 131 ± 13 vs. 128 ± 9 mmHg; HDG: −1%, 128 ± 9 vs. 128 ± 7 mmHg). 

*Diastolic Blood Pressure (DBP):* There was no significant interaction for group x time (*p* = 0.424) or main effect for group (*p* = 0.979). There was a significant main effect for time (*p* = 0.002). At 0POST, DBP decreased by of 9% and 5% in LDG (84 ± 12 vs. 77 ± 8 mmHg, *p* = 0.006, d = 0.7) and HDG (83 ± 8 vs. 77 ± 6 mmHg, *p* = 0.02, d = 0.9), respectively. At 3POST, there was no significant difference in DBP in either group (LDG: −1%, 77 ± 8 vs. 76 ± 11 mmHg; HDG: +4%, 77 ± 6 vs. 80 ± 8 mmHg). 

## 4. Discussion

Despite the increased popularity of SIT exercise paradigms due to purported physiological improvements and health benefits, there is a paucity of research that has investigated the dose-response effects of concurrent SIT/RT programs in previously inactive populations. Whilst a considerably reduced volume of concurrent SIT/RT has been shown to induce similar physiological adaptations compared to a high-volume dose after 6 wks using 20 s SIT bouts [11], an extended post intervention period has not been investigated. By investigating the dose response at multiple time points post intervention, it could help provide important information regarding the time course of changes in the physiological and health adaptations acquired. In addition, the tolerability of introducing longer duration bouts of SIT in previously inactive populations may be an issue. Therefore, we investigated the dose-response of two 6 wk short-duration bout SIT/RT protocols on indices of aerobic capacity and health, including after a 3 wk training cessation period. We found that: 1) At 0POST, both the LDG and HDG group significantly increased VO_2peak_ by 16% and 11%; 2) at 3POST, VO_2peak_ decreased (−11%) in the LDG group, but was maintained (+3%) in the HDG group; and 3) PE increased and RPE decreased over the 6 wk intervention in both groups, suggesting that sessions became more enjoyable and tolerable, which has potential positive implications for longer-term adherence. 

*Cardiorespiratory fitness:* A significant increase in VO_2peak_ for both groups was expected due to the untrained status of the participants [21]. Repeated ‘all-out’ bouts have been shown to improve VO_2peak_ [22], with a cumulative total of 1 min SIT protocols increasing VO_2peak_ in overweight populations [23]. The magnitude of improvements in VO_2peak_ in our study is similar to the 15% and 12% increases using 20 s SIT protocols in overweight men and women (3/wk for 6 wks) by Metcalfe et al. [3] and Gillen et al. [7], respectively. The mechanisms responsible for increases in VO_2peak_ following SIT are unclear, but likely reflect central and peripheral adaptations [24]. It must also be noted that the mean body mass index (BMI) of the groups reported in the Metcalfe et al. [3] study are lower than both the study by Gillen et al. [7] and our study. It is not known whether these differences had any effect on the magnitudes of findings.

There was no significant difference in VO_2peak_ values between the LDG (+16%) and HDG (+11%) groups at 0POST, albeit a trend for a greater magnitude in the LDG group. Logan et al. [11] found that the greatest change in VO_2peak_ amongst five groups occurred in the group who completed the lowest volume of training. However, the divergence between groups at 3POST in our study suggests that aerobic capacity in the HDG group may have been suppressed at 0POST. At 3POST, the 11% decrease in VO_2peak_ in the LDG was expected owing to detraining effects [25]. However, VO_2peak_ was maintained (+3%) in the HDG group. Although speculative, these findings suggest temporary overreaching and a possible delayed training effect in the HDG group. Athletic populations deliberately invoke temporary overreaching after an intense block of periodised training, which transiently compromises performance [26,27]. The benefits are normally only fully realized once a strategic reduction in training load (tapering) is implemented, which reverses functional overreaching (residual fatigue) and elevates performance to a ‘super-compensated’ level [28,29]. In sedentary populations, high volumes of repeated maximal efforts over prolonged time periods (weeks) can also invoke overreaching [29]. It is plausible that a temporary state of overreaching was induced in the HDG group at 0POST, which was absent in the LDG group due to the considerably reduced volume. It is entirely likely the HDG group experienced similar detraining effects to the LDG group. However, we speculate that the detraining effects were masked more in the HDG group due to super-compensation of training helping to ‘offset’ the detraining effects. In support, Hatle et al. [30] showed that following a high frequency treadmill running training protocol (4 × 4 min @ 90%–95% HR_MAX_: 24 sessions over 3 wks), aerobic adaptations were depressed, but increased by 6% following a 12-day detraining period. Although the phenomenon of delayed training effects is known to occur, the underpinning mechanisms underlying the time course kinetics are currently not well understood. More research is needed, as there currently is a paucity of research in training intervention studies that investigate an extended post intervention period (<72 h).

*Blood Pressure:* Our findings of no significant changes in SBP for either group at 0POST are similar to Rakobowchuk et al. [31], who also reported no change in SBP following 6 wks of SIT. However, Whyte et al. [32] reported a significant 4.7% reduction in SBP in sedentary normotensive men following 2 wks of cycle based SIT (4–6 × 30 s:4 min, 3×/wk). Similarly, Adamson et al. [33] reported a 9% reduction in SBP post SIT (10 × 6 s:1 min @ 3×/wk for 6 wks). These studies used more ‘hypertensive’ participants, which may help explain the findings, as resting BP levels are purported to be a major predictor of the magnitude of change [34]. There were significant reductions of 9% and 5% in DBP in the LDG and HDG groups, respectively, at 0POST, which did not change significantly at 3POST. Similarly, Bonsu and Terblanche [18] reported a 4.3 mmHg reduction in DBP following 2 wks of HIIT, which was sustained following a 2 wk detraining period. Logan et al. [11] found an average decrease of 7% across their treatment groups. In contrast to our study, the highest reductions were seen in their highest volume groups, 4 (−13%) and 5 (−10%), which may be expected as higher volumes of training are associated with larger reductions in DBP [35]. The significant reductions in DBP are likely beneficial, especially as blood pressure increases throughout adult life, increasing the chances of hypertension [18].

*Blood Lipids:* The significant reductions of 16% and 7% in TC in the LDG and HDG groups, respectively, at 0POST, are similar to Fisher et al. [36], who found a reduction in TC (9.2%) following 6 wks of 3×/wk HIIT (4 × 30 s @ 85% peak power output (PPO)) with 4 min @ 15% PPO). Whereas, we found no significant differences for TG, they showed a significant reduction (15.3%). The reasons for our TG findings, or why the magnitude of TC changes is greater in the LDG group, are unclear. However, as alterations in the blood lipid profile appear to be influenced by overall energy expenditure [37], perhaps factors, such as energy intake, contributed, especially as SIT can suppress appetite immediately post exercise and affect daily energy intake [38]. Future studies should impose tighter control of these potentially lipid altering factors. Also, why the findings for both TC and TG at BL2 significantly changed from BL1 is unclear, especially as testing procedures were very tightly controlled and all other variables remained stable during the control period.

*Psychological Responses:* Shorter duration SIT bouts are perceived as being more enjoyable and tolerable than longer duration bouts [39], whilst also likely resulting in less severe feelings of nausea or fatigue [40]. That said, it was still expected that the initial exposure to such an intense stimulus in previously inactive participants would induce considerable physical and psychological distress both during and immediately after sessions, particularly in the HDG group. The findings of significant increases in session PE and decreases in session RPE in both groups, with, surprisingly, no significant differences between groups, are important from a practical perspective. It would be prudent for practitioners to emphasize the implementation of progressive training structures, particularly in the early stages, by incorporating planned periods of appropriate rest and recovery. This may help participants successfully tolerate the crucial initial stages of SIT participation, where attrition rates are likely highest. Improved PE and RPE may help promote long-term adherence, but this is speculative at present. It is notable that of the six participants who withdrew from this study, none cited intolerability to the protocol as the primary reason. 

*Study Limitations:* When interpreting these findings, it needs to be considered that the sample size is small. This was unfortunately due to some late participant withdrawals (overall attrition of ~30%). This attrition also affected the anthropometric matching of the groups for BM and %BF, as two of the participants randomized to the HDG group who withdrew had higher BM. There is also a possibility that the non-significant differences between groups for BM may have been influenced by the combination of a large SD for the LDG group and the small sample size. As discussed previously, whether differences in BM would have any effect on the findings is unknown and should be investigated in future studies. These findings also suggest that temporary overreaching may have been implicated. However, this is currently speculative, and future studies using larger sample sizes should be conducted to investigate this hypothesis further and to try and establish physiological mechanisms.

## 5. Conclusions

We hypothesized that the dose response of a concurrent SIT/RT intervention in a previously inactive population may affect the time course of aerobic adaptations post intervention. The main findings of significantly increased VO_2peak_ at 0POST for both the LDG and HDG groups with no differences between groups was expected. However, the findings at 3POST suggest that VO_2peak_ may have been compromised at 0POST in the HDG group. We speculate that this may have been due to a temporary state of overreaching in the HDG group that masked the full magnitude of adaptations, which were not realized until 3POST, when they appeared to help offset the expected detraining effects. These novel findings highlight the importance of sustained exercise training for improving or maintaining acquired health adaptations and should be considered by practitioners when implementing structured concurrent SIT/RT programs in non-athletic populations, especially when using high-dose protocols. Our findings suggest that there may be a risk of overreaching when cumulative programs are administered to previously inactive individuals without strategic periods of tapering incorporated. Also important, the similar psychological responses between groups suggests that higher dose short-bout SIT protocols are well tolerated in previously inactive populations.

## Figures and Tables

**Figure 1 sports-07-00085-f001:**
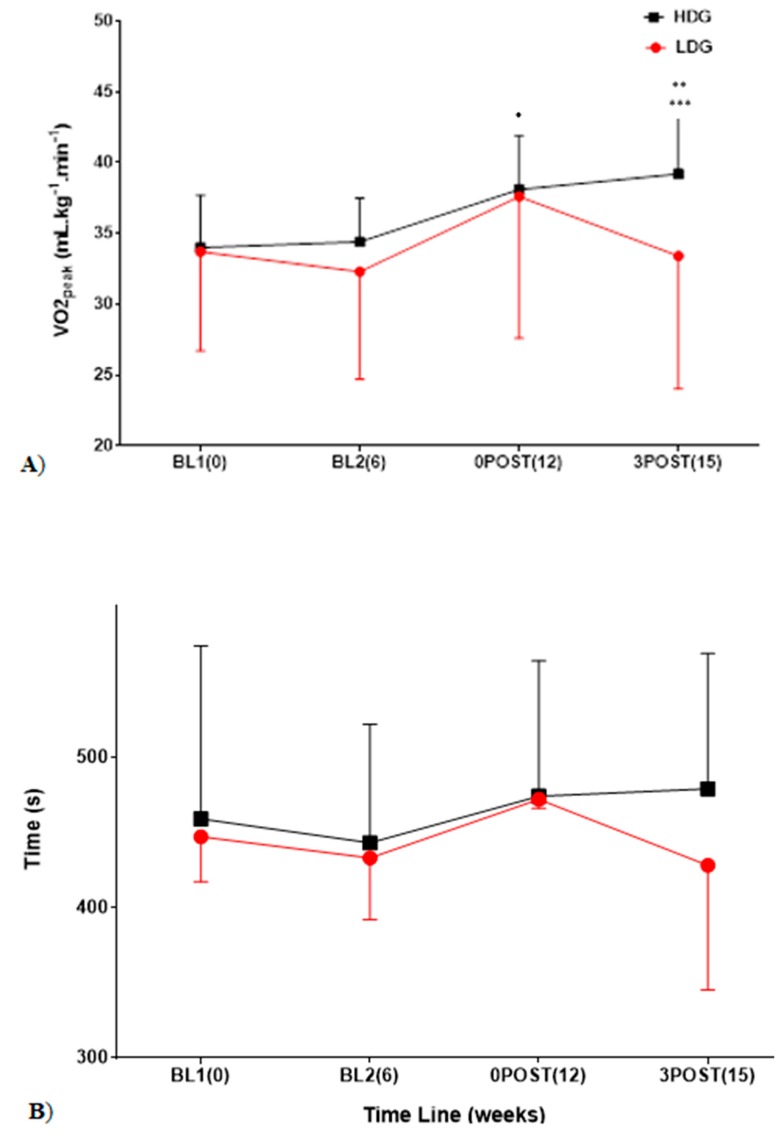
Time course of changes in VO_2peak_ (**A**) and time to exhaustion (**B**). * Indicates *p* < 0.05 for main effect of time. ** Indicates *p* < 0.05 for main effect of group ^***^ indicates *p* < 0.05 for interaction group × time.

**Figure 2 sports-07-00085-f002:**
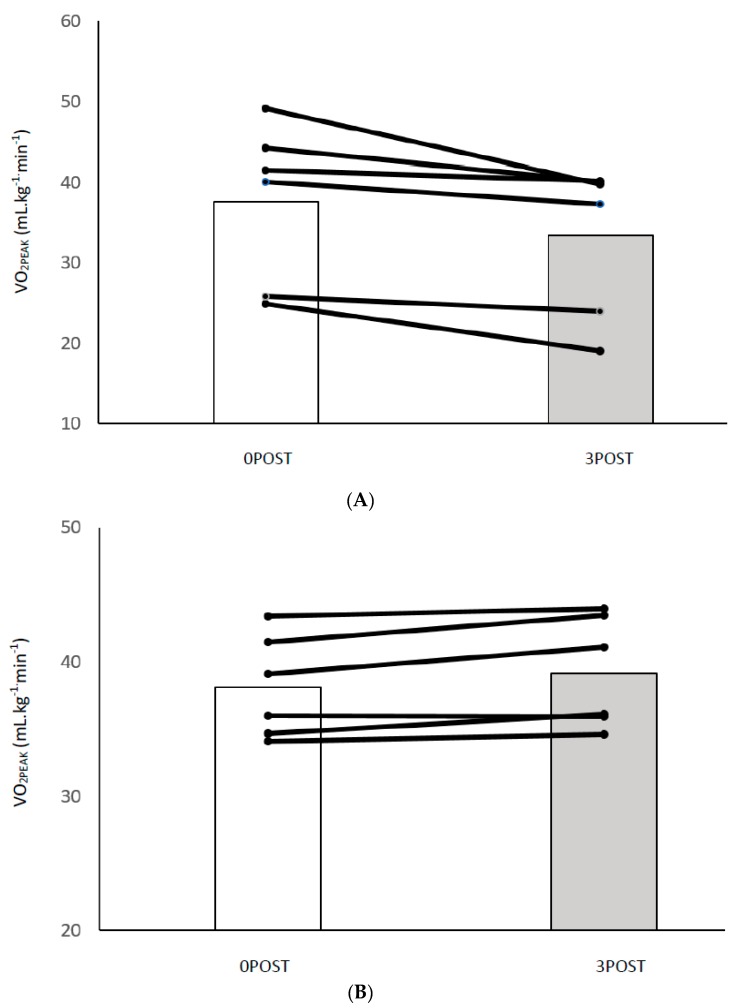
VO_2PEAK_ individual data for LDG (**A**) and HDG (**B**) for 0POST vs. 3POST.

**Figure 3 sports-07-00085-f003:**
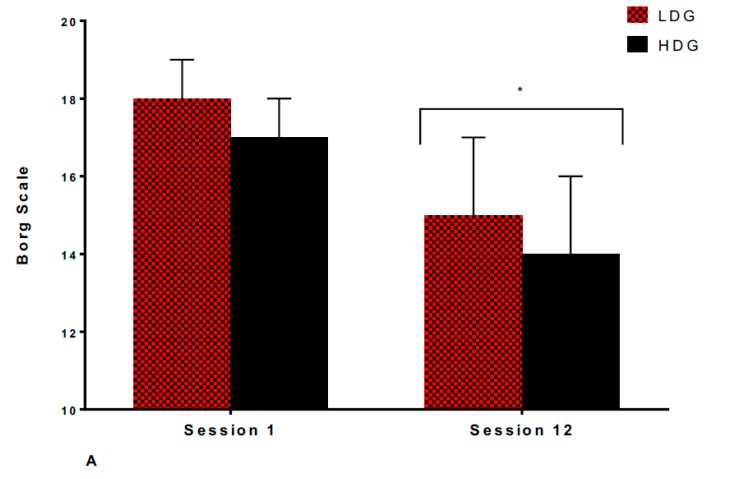
A comparison of **A**) RPE; **B**) PE for LDG vs. HDG during SIT sessions 1 and 12. Data is shown as M ± SD. * Significantly different to session 1 (*p* < 0.05).

**Table 1 sports-07-00085-t001:** A comparison of anthropometric indices for LDG vs. HDG across time points. Data is shown as M ± SD. * Significantly different to BL2 (*p* < 0.05).

	Group	BL1	BL2	0POST	3POST	Time	Group × Time
Body Mass (kg)	LDG	95.4 ± 27.8	96.0 ± 26.5	96.0 ± 26.1	96.0 ± 26.9	*p* = 0.987	*p* = 0.622
	HDG	86.4 ± 7.0	85.8 ± 7.4	85.8 ± 6.5	86.1 ± 7.5		
Body Fat (%)	LDG	27.6 ± 7.8	27.4 ± 7.6	27.3 ± 7.5	27.8 ± 7.3	*p* = 0.195	*p* = 0.463
	HDG	24.6 ± 2.5	24.8 ± 2.8	23.8 ± 2.6	24.8 ± 2.7		
Muscle Mass (kg)	LDG	69.0 ± 7.4	69.0 ± 7.2	69.0 ± 7.0	68.6 ± 6.9	*p* = 0.260	*p* = 0.468
	HDG	71.7 ± 2.4	71.5 ± 2.7	72.4 ± 2.5	71.7 ± 2.5		
Trunk Fat (%)	LDG	31.1 ± 7.5	31.3 ± 7.6	31.4 ± 7.4	30.0 ± 6.7	*p* = 0.584	*p* = 0.125
	HDG	27.3 ± 3.2	27.5 ± 3.4	26.3 ± 3.1	27.2 ± 1.9		
Right Leg Girth (cm)	LDG	61 ± 8.0	62 ± 6.0	65 ± 6.0 *	64 ± 6	*p* < 0.001	*p* = 0.158
	HDG	58 ± 2.0	60 ± 3.0	60 ± 3.0	60 ± 3.0		
Right Leg Muscle (%)	LDG	11.5 ± 1.9	11.5 ± 1.9	11.6 ± 1.9	11.4 ± 1.9	*p* = 0.540	*p* = 0.137
	HDG	10.8 ± 0.9	10.6 ± 0.9	10.6 ± 0.8	10.6 ± 1.0

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
