# Peer review of "The Dose Response for Sprint Interval Training Interventions May Affect the Time Course of Aerobic Training Adaptations"

_sports, 2019, doi:10.3390/sports7040085_

Round 1

Reviewer 1 Report

This manuscript sought to investigate the effects of concurrent training (SIT/RT) in a low dose and a high dose group over 6-weeks, and then after a 3-week detraining period.  This manuscript is well-written, and offers a very unique hypothesis. It's strength is in it's design.  However, there are issues that concern me.

Major:

-Is six subjects per group enough power to make any assumptions about these data?  No power equation is provided, and I suspect that this study is under-powered.

-The LDG is classified as overweight, while the HDG is obese.  The SD for body mass in the LDG is 3 times that of the HDG, but that is never addressed.  I suspect that due to this, the groups were statistically the same. But, again, with only six subjects these data are limited in their interpretation.  Do the authors feel that the differences in BMI, and even BM, between these groups, while not statistically significant, played a role in the responses?  I suspect that they may have, but I saw no mention of it whatsoever.  I think the authors have to discuss these differences, instead of shying away from them.

Minor:

-PAge 2, line 88. "Females...', best to use 'women', as you used 'men' earlier.

-Page 2, line 90. Did you control for any medications?

-Page 4, line 165. Where any of the data not normally distributed? Again, with six subjects per group, I don't see how they could be normally distributed. 

-PAge 4, line 166. Is it appropriate to use an ANOVA on RPE?  It is not a continuous variable.  Please explain why this decision was made.  In addition, with only 6 subjects is an ANOVA appropriate at all?  Also, the authors mention Cohen's d here, but I didn't that used anywhere in the Results.

-Page 5, line 176. The authors repeat %change, but that was not in the Methods section.  Did they authors analyze %change, or just report it?  Also, why was the decision made to just report p values?  Alpha levels don't provide information on strength of relationship, like Cohen's d, just probability of error. 

-Page 10, line 261.  How do the authors justify comparing their data to overweight subjects (Metcalfe et al.) when they had one group of obese subjects? Do you think this altered your data?  I find this an important discussion point.

-Page 10, line 282. What was the 'high frequency training'?   

Author Response

Dear Reviewer,

We thank you for your comments.  They have been very helpful to help improve the quality of the manuscript. We have addressed your comments point-by-point below, We welcome any feedback on these changes.

-Is six subjects per group enough power to make any assumptions about these data?  No power equation is provided, and I suspect that this study is under-powered.

-The LDG is classified as overweight, while the HDG is obese.  The SD for body mass in the LDG is 3 times that of the HDG, but that is never addressed.  I suspect that due to this, the groups were statistically the same. But, again, with only six subjects these data are limited in their interpretation.  Do the authors feel that the differences in BMI, and even BM, between these groups, while not statistically significant, played a role in the responses?  I suspect that they may have, but I saw no mention of it whatsoever.  I think the authors have to discuss these differences, instead of shying away from them.

Response: thank you for the above comments.  We have included a study limitations section (Line 342-351) where we have discussed your concerns and presented some suggestions for future research too. We were frustrated to have had an attrition of over 30% and as a result the final sample size is quite small. We included effect sizes and the presentation of individual data points for the VO2 data but do agree that this needs to be recognised formally in the manuscript.

Minor:

-PAge 2, line 88. "Females...', best to use 'women', as you used 'men' earlier.

Response: the inconsistency was an oversight, thank you for pointing this out.  We have changed to ‘male’ instead of ‘men’ on line 81

-Page 2, line 90. Did you control for any medications?

Response: yes medications were controlled, no patients included in the study were on any medications.

-Page 4, line 165. Where any of the data not normally distributed? Again, with six subjects per group, I don't see how they could be normally distributed. 

Response:  data were analysed for normality using Shapiro-Wilks, which reported normal distribution of the data set.

-PAge 4, line 166. Is it appropriate to use an ANOVA on RPE?  It is not a continuous variable.  Please explain why this decision was made.  In addition, with only 6 subjects is an ANOVA appropriate at all?  Also, the authors mention Cohen's d here, but I didn't that used anywhere in the Results.

Response:  thank you highlighting this. We have re- run our analysis to look at group differences in RPE and PE using an independent sample t-test. Within group differences were analysed via paired samples t-test. This has been added to the methods and results have been updated (P-values have changed very slightly, and the overall outcomes remain the same).

-Page 5, line 176. The authors repeat %change, but that was not in the Methods section.  Did they authors analyze %change, or just report it?  Also, why was the decision made to just report p values?  Alpha levels don't provide information on strength of relationship, like Cohen's d, just probability of error. 

Response: thank you for the comment, % change was not analysed, only reported within the manuscript. Cohens d has been added to RPE and PE results.

-Page 10, line 261.  How do the authors justify comparing their data to overweight subjects (Metcalfe et al.) when they had one group of obese subjects? Do you think this altered your data?  I find this an important discussion point.

Response: thank you for the comment.  The BMIs reported in the Metcalfe study are lower than both our study and the Gillen study.  We have now included this in the discussion (L274-276) and also in the study limitations section (L343-352).

-Page 10, line 282. What was the 'high frequency training'?   

Response: thank you, this is clarified now on Line 293

Reviewer 2 Report

Title:  recommend "dose" be capitalized as "Dose".

Abstract: 

Line 9 - Define SIT (and all other acronyms) at first use.

Line 10 - Remove comma after "(<10s)".  I only make this comment as an example of the requirement to review the entire manuscript and edit for typographical, grammatical, and other unintended errors.

Introduction:

Line 30 - Define "uptake".

Methods:

Line 164 - did you perform a priori power analysis to determine the sample size appropriate for your design and dependent variables?  If so, please report.  If not, your presentation of effect sizes along with participant drop out seems adequate.

Results:

Line 174 - sentence appears incomplete.

Discussion:

Line 241 - first sentence seems to run on; recommend editing to simplify.

Line 254 - while most will agree that PE "improved", I do not agree that a decrease in RPE should necessarily be classified as "improved".  Recommend clarification of terminology.

While the detraining aspect and explanation of overreaching are important, the inclusion of resistance training without dependent variables and lack of alternative explanations of findings do require expanded discussion.  Please explain why you included RT but did not assess any change in related variables.

Author Response

Dear Reviewer,

We thank you for your comments.  They have been very helpful to help improve the quality of the manuscript. We have addressed your comments point-by-point below, We welcome any feedback on these changes.

Title:  recommend "dose" be capitalized as "Dose".

Response:  based on the recommendation of another reviewer, we have changed the title to better reflect the study.

Abstract: 

Line 9 - Define SIT (and all other acronyms) at first use.

Line 10 - Remove comma after "(<10s)".  I only make this comment as an example of the requirement to review the entire manuscript and edit for typographical, grammatical, and other unintended errors.

Response: thank you for your comments regarding the abstract. Issues with the clarity of the abstract were also highlighted by other reviewers. Therefore, we have substantially restructured the abstract to make it more succinct and provide for better clarity and hope it reads better now.  We welcome any feedback on our amendments.

Introduction:

Line 30 - Define "uptake".

Response: thank you, this has been amended (L 30)

Methods:

Line 164 - did you perform a priori power analysis to determine the sample size appropriate for your design and dependent variables?  If so, please report.  If not, your presentation of effect sizes along with participant drop out seems adequate.

Response: thank you for your comment.  This issue was also highlighted by other reviewers. As a result, we have discussed your concerns, and presented some suggestions for future research too. We were frustrated to have had an attrition of over 30% and as a result the final sample size is quite small. We included effect sizes and the presentation of individual data points for the VO2 data.

Results:

Line 174 - sentence appears incomplete.

Response: thank you, the words have been deleted

Discussion:

Line 241 - first sentence seems to run on; recommend editing to simplify.

Response: thank you, this sentence has been amended (L252-254)

Line 254 - while most will agree that PE "improved", I do not agree that a decrease in RPE should necessarily be classified as "improved".  Recommend clarification of terminology.

Response: thank you, this statement has been amended for better clarity (L 264-266)

While the detraining aspect and explanation of overreaching are important, the inclusion of resistance training without dependent variables and lack of alternative explanations of findings do require expanded discussion.  Please explain why you included RT but did not assess any change in related variables.

Response:  thank you for your comment here.  We included RT as it reflects how RT is now recommended to be integrated within most exercise training programmes. For example, the ACSM amended their guidelines to incorporate RT as an integral part of all training programmes. Therefore, we wanted to include a protocol that would reflect the common practice and recommendations for incorporating RT within training interventions for previously inactive populations, therefore increasing its ecological validity. We touch on this in the introduction (L58-62)

Reviewer 3 Report

Comments:

1.     Title: too general. Not suitable for this manuscript. It would be good for review articles. Title needs to be changed.

2.     Abstract:  should be understandable for a person who does not know the main text. I do not think that  is this  case. Abstract is unreadable. There are unexplained shortcuts make the text incomprehensible to a reader who does not know the main text (eg. BL 1).

Line 22 - "Our results have important implications...". The Authors have only studied 13 people in two groups (6+7; inactive populations). This does not authorize such a statement.

3.     Introduction:

  line 32 – „moderate intensity interval training (MICT)”. Do you use in further text this abbreviation? If not, then there is no need to give a shortcut.

Line 69 -  “post-intervention but;”. Why did you write “but”?

- line 65 - "RT" - it should be explain. It is explained only in line 111.

4.         Materia and method: very small research group. I have doubts whether such a small research group is sufficient to draw general conclusions. Such a small group would be justified when researching, for example, some unique group of people (eg. athletes - representatives of the country).

 Material and Method (and in other places) – bad bibliography entries. Line 108 -  “. [16]”; it should be “[16].”.  Compare with line 48.

-          Table. 1 – most of data is in Table. 2 (body mass, body fat). Do not repeat the data. I suggest you give up table 1.

-          Line 100 „A minimum of 48 h separated sessions”. This sentence is not clear.

-          Line 74 (and in other places) You use the sign: @ , & . Why? What does it mean?

-          Line 125 “24h”, line 127 “24 h”. Why is differently?

-          Line 150-151 – It should be explained what does mean: CHO, FAT, PRO.

5.         Results

-          Line 173 – “Fig. 1a”, rather “Fig. 1A”. See figures (A, B, not a,b).

-          Fig. 3B session 12 - Can you check if there is a statistically significant difference?

-          Line 207 – “(p=0.097), There were”; Why “There”, but not “there”.

I am interested in how exactly the six-second effort was measured on the cycle ergometer (point 2.4).

6.         Discussion the last paragraph is missing a summary of the discussion - no reference to hypotheses (not yet the conclusion).

- line 241-250 - it is rather "an Introduction"

- line 250-255 - it is rather "Results"

- line 255-256 - it is rather "Conclusions"

7.         Conclusion – line 326-330 – It is rather a discussion.

8. References:

References have been entered contrary to the editorial requirements.

See the pattern on the website.

- No 22 - from 1996 - is necessary? Current? Maybe are there newer research results on this subject?

- No 29 - "31  44-46" - Is it correct? Please, check it.

Author Response

Many thanks for your comments.  They have been very helpful to help improve the quality of the manuscript. We have addressed your comments point-by-point below, We welcome any feedback on these changes.

Comments:

1.     Title: too general. Not suitable for this manuscript. It would be good for review articles. Title needs to be changed.

Response: thank you for your comment here. On reflection, we agree that the title would be more suited to a review paper.  Therefore, we have changed the title to better reflect the current study

2.     Abstract:  should be understandable for a person who does not know the main text. I do not think that  is this  case. Abstract is unreadable. There are unexplained shortcuts make the text incomprehensible to a reader who does not know the main text (eg. BL 1).

Line 22 - "Our results have important implications...". The Authors have only studied 13 people in two groups (6+7; inactive populations). This does not authorize such a statement.

Response: thank you for your comments regarding the abstract. Issues with the clarity of the abstract were also highlighted by other reviewers. Therefore, we have substantially restructured the abstract to make it more succinct and provide for better clarity and hope it reads better now.  We welcome any feedback on our amendments.

3.     Introduction:

  line 32 – „moderate intensity interval training (MICT)”. Do you use in further text this abbreviation? If not, then there is no need to give a shortcut.

Response: thank you, we use it again on L46, which is why we abbreviated.

Line 69 -  “post-intervention but;”. Why did you write “but”?

Response: thank you, we included it to link the two points together, however, we have removed it.

- line 65 - "RT" - it should be explain. It is explained only in line 111.

Response: thank you, this has been amended, and appears first in L60.

4.         Materia and method: very small research group. I have doubts whether such a small research group is sufficient to draw general conclusions. Such a small group would be justified when researching, for example, some unique group of people (eg. athletes - representatives of the country).

Response: thank you for the comment, we have included a study limitations section to our discussion which addresses this concern (Line 343-352)

 Material and Method (and in other places) – bad bibliography entries. Line 108 -  “. [16]”; it should be “[16].”.  Compare with line 48.

Response: thank you, this has been amended throughout the manuscript.

-          Table. 1 – most of data is in Table. 2 (body mass, body fat). Do not repeat the data. I suggest you give up table 1.

Response: thank you for this suggestion, we agree with you here and have omitted Table 1 now.  We have included the VO2 data into the main text on L181-183

-          Line 100 „A minimum of 48 h separated sessions”. This sentence is not clear.

Response: thank you, this has been edited for better clarity (L99)

Line 74 (and in other places) You use the sign: @ , & . Why? What does it mean?

Response: thank you, we used @ to denote ‘at’ as it is commonly used. ‘&’ is used as a shortened version of ‘and’

Line 125 “24h”, line 127 “24 h”. Why is differently?

Response: thank you for flagging this, is a typo and has been amended

Line 150-151 – It should be explained what does mean: CHO, FAT, PRO.

Response: thank you, these were used to denote carbohydrate, fat and protein content, and has been amended (L154-155)

5. Results

Line 173 – “Fig. 1a”, rather “Fig. 1A”. See figures (A, B, not a,b).

Response: thank you, these have been amended

Fig. 3B session 12 - Can you check if there is a statistically significant difference?

Response: thank you, yes there is a significant difference for time

Line 207 – “(p=0.097), There were”; Why “There”, but not “there”.

Response: thank you, this has been amended

I am interested in how exactly the six-second effort was measured on the cycle ergometer (point 2.4).

Response: sprints were carried out on a manually loaded cycle ergometer. Sprints began when the participant exceeded 100 RPM, whereby the resistance automatically dropped. Each sprint was timed by the investigator with the resistance removed manually by the investigator at the end of each 6 s sprint. Each sprint was counted down during 18 s recovery period.

6.  Discussion the last paragraph is missing a summary of the discussion - no reference to hypotheses (not yet the conclusion).

- line 241-250 - it is rather "an Introduction"

Response: thank you for the comments here.  We used this format as we felt it is common at the start of many discussions to provide an executive summary (including rationale for the study, aims, main findings and inference) to set a blueprint for the discussion. We have amended the conclusion (L354-367). However, we are open to suggestions here and welcome any feedback or suggestions.

- line 250-255 - it is rather "Results"

Response: as above

- line 255-256 - it is rather "Conclusions"

Response: thank you, we have removed this line from this part of the discussion

7. Conclusion – line 326-330 – It is rather a discussion.

Response: thank you, we used this format as we felt it included a concise summary of the main findings and also provided an inference.  However, as above we would welcome any feedback or suggestions here.

8.  References:

References have been entered contrary to the editorial requirements.

See the pattern on the website.

Response: thank you for flagging this up. References now updated to reflect the journal requirements.

- No 22 - from 1996 - is necessary? Current? Maybe are there newer research results on this subject?

Response: thank you for your comment.  We included this as we felt that is was a well-controlled study and the results were relevant to our discussion, even if it is quite old. We have also backed up the point with recent studies.

- No 29 - "31  44-46" - Is it correct? Please, check it.

Response: this reference has been amended, thank you.

Reviewer 4 Report

The authors presented the study in a clear and straightforward manner.  Background information is sufficient. Results are demonstrated well with appropriate tables and figures.  Study was well controlled with sound scientific designs.  Speculated mechanisms were provided for the discussion.  Overall, the writing is intriguing.

Line 9-23. Extensive acronyms were used to make it difficult to grasp the meaning of the terms for the first-time viewers.  Consider using more reader friendly format (e.g, full term) before adopting the acronyms.

Line 17. “Perceived Enjoyment and RPE increased and decreased for both groups from Session 1 to 12 of intervention.”  Consider rewrite for clarifications.

Line 39-40.  Why is “making them 40 potentially more” having a different font size than the rest of the text?

Line 84-85.  Please elucidate the research design using “random, stratified approach based on baseline VO2peak.” How VO2peak was stratified.  

Line 94.  Although it  appears that Control Period “Began immediately after BL1….”, the first sentence at Line 94 seems missing a subject.  Consider editing.

Line 111.  It’s not clear of the subject(s) in the sentence “Participants 10 repetition maximum (R-M) ….”.    Is it participants or 10 repetition maxima (R-M)?

Line 174. What’s the last word “there” referring to in the sentence “ main effect of group….”.  It seems a incomplete sentence.

Line 183. Is there a unit assigned to “time to exhaustion”?  If so, please include in the sentence and the rest of the paragraph.

Line 211. Is the second “significant” supposed to be “significantly”?

Line 212. The first word “And” needs to be the lowercase.

Line 224-229. It is suggest Table 2 be eliminated as no major significant except the one time point.  The data report can be descriptive.

Line 232.  Please clean up the space and punctuations issues in the first sentence.  

Line 237-239. What is the measurement unit for BP results to be added?

The discussion: This section left readers wondering the comparable studies on dose-response relationship using SIT in a HIIT perspective.  More information on the comparisons could be included but acceptable with current version.  It’s suggested that the authors discuss the strengths and weaknesses of the current study to shed lights for future studies.      

Author Response

Many thanks for your comments.  They have been very helpful to help improve the quality of the manuscript. We have addressed your comments point-by-point below, We welcome any feedback on these changes.

Line 9-23. Extensive acronyms were used to make it difficult to grasp the meaning of the terms for the first-time viewers.  Consider using more reader friendly format (e.g, full term) before adopting the acronyms. Line 17. “Perceived Enjoyment and RPE increased and decreased for both groups from Session 1 to 12 of intervention.”  Consider rewrite for clarifications.

Response: thank you for your comments regarding the abstract. Issues with the clarity of the abstract were also highlighted by other reviewers. Therefore, we have substantially restructured the abstract to make it more succinct and provide for better clarity and hope it reads better now.  We welcome any feedback on our amendments.

Line 39-40.  Why is “making them 40 potentially more” having a different font size than the rest of the text?

Response: thank you, this has been amended

Line 84-85.  Please elucidate the research design using “random, stratified approach based on baseline VO2peak.” How VO2peak was stratified.  

Response: thank you, participants were matched by VO2peak and randomly assigned to LDG or HDG.

Line 94.  Although it appears that Control Period “Began immediately after BL1….”, the first sentence at Line 94 seems missing a subject.  Consider editing.

Response: thank you, this has been amended.

Line 111.  It’s not clear of the subject(s) in the sentence “Participants 10 repetition maximum (R-M) ….”.    Is it participants or 10 repetition maxima (R-M)?

Response: thank you, this has been amended.

Line 174. What’s the last word “there” referring to in the sentence “ main effect of group….”.  It seems a incomplete sentence.

Response: thank you, this has been amended.

Line 183. Is there a unit assigned to “time to exhaustion”?  If so, please include in the sentence and the rest of the paragraph.

Response: thank you, yes TTE unit is seconds(s) and this has been amended (L190-195)

Line 211. Is the second “significant” supposed to be “significantly”?

Response: thank you, this has been amended.

Line 212. The first word “And” needs to be the lowercase.

Response: thank you, this has been amended.

Line 224-229. It is suggest Table 2 be eliminated as no major significant except the one time point.  The data report can be descriptive.

Response: thank you for this suggestion, we agree with you here and have omitted Table 1 now.  We have included the VO2 data into the main text on L181-183

Line 232.  Please clean up the space and punctuations issues in the first sentence.  

Response: thank you, this has been amended.

Line 237-239. What is the measurement unit for BP results to be added?

Response: thank you, this has been amended (L241-250)

The discussion: This section left readers wondering the comparable studies on dose-response relationship using SIT in a HIIT perspective.  More information on the comparisons could be included but acceptable with current version.  It’s suggested that the authors discuss the strengths and weaknesses of the current study to shed lights for future studies.

Response:  thank you for the comment. We have included a study limitations section (L342-353) where we have discussed your concerns and presented some suggestions for future research too.

Round 2

Reviewer 1 Report

The revised manuscript is a significant improvement over the original, and the authors should be congratulated.  However:

    -The caption for Fig 1 doesn't match the text. The text states, 'group by time (p=0.013), group (p<0.001) and time (p<0.001). The graph just says p<0.05, with no mention of main effect of group whatsoever. The same issue with the p values also plagues Fig 3.

    -I can't read Table 1, it is offset.

Author Response

The revised manuscript is a significant improvement over the original, and the authors should be congratulated.  However:

    -The caption for Fig 1 doesn't match the text. The text states, 'group by time (p=0.013), group (p<0.001) and time (p<0.001). The graph just says p<0.05, with no mention of main effect of group whatsoever. The same issue with the p values also plagues Fig 3.

Response: Thank you. Figure captions have been updated to match the text within the results.

    -I can't read Table 1, it is offset.

Response: Thank you. Table 1 has been edited and aligned. 

Reviewer 3 Report

Thank you for ypur responce.

Author Response

Thank you for your valuable reviews.